# Peto’s “Paradox” and Six Degrees of Cancer Prevalence

**DOI:** 10.3390/cells13020197

**Published:** 2024-01-21

**Authors:** Andras Szasz

**Affiliations:** Department of Biotechnics, Hungarian University of Agriculture and Life Sciences, 2100 Gödöllő, Hungary; biotech@gek.szie.hu

**Keywords:** Peto’s paradox, mutations, six degrees of stages, evolutional game, Darwin’s law, allometry

## Abstract

Peto’s paradox and the epidemiologic observation of the average six degrees of tumor prevalence are studied and hypothetically solved. A simple consideration, Petho’s paradox challenges our intuitive understanding of cancer risk and prevalence. Our simple consideration is that the more a cell divides, the higher the chance of acquiring cancerous mutations, and so the larger or longer-lived organisms have more cells and undergo more cell divisions over their lifetime, expecting to have a higher risk of developing cancer. Paradoxically, it is not supported by the observations. The allometric scaling of species could answer the Peto paradox. Another paradoxical human epidemiology observation in six average mutations is necessary for cancer prevalence, despite the random expectations of the tumor causes. To solve this challenge, game theory could be applied. The inherited and random DNA mutations in the replication process nonlinearly drive cancer development. The statistical variance concept does not reasonably describe tumor development. Instead, the Darwinian natural selection principle is applied. The mutations in the healthy organism’s cellular population can serve the species’ evolutionary adaptation by the selective pressure of the circumstances. Still, some cells collect multiple uncorrected mutations, adapt to the extreme stress in the stromal environment, and develop subclinical phases of cancer in the individual. This process needs extensive subsequent DNA replications to heritage and collect additional mutations, which are only marginal alone. Still, together, they are preparing for the first stage of the precancerous condition. In the second stage, when one of the caretaker genes is accidentally mutated, the caused genetic instability prepares the cell to fight for its survival and avoid apoptosis. This can be described as a competitive game. In the third stage, the precancerous cell develops uncontrolled proliferation with the damaged gatekeeper gene and forces the new game strategy with binary cooperation with stromal cells for alimentation. In the fourth stage, the starving conditions cause a game change again, starting a cooperative game, where the malignant cells cooperate and force the cooperation of the stromal host, too. In the fifth stage, the resetting of homeostasis finishes the subclinical stage, and in the fifth stage, the clinical phase starts. The prevention of the development of mutated cells is more complex than averting exposure to mutagens from the environment throughout the organism’s lifetime. Mutagenic exposure can increase the otherwise random imperfect DNA reproduction, increasing the likelihood of cancer development, but mutations exist. Toxic exposure is more challenging; it may select the tolerant cells on this particular toxic stress, so these mutations have more facility to avoid apoptosis in otherwise collected random mutational states.

## 1. Introduction

We are in a war against cancer. The battle extended a wide area of medicine and impacted society. The results must be won or lost. The situation is more complex. The “win” is the falling mortality and suppressed cancer morbidity. The rapid growth of the worldwide population and the intensive aging are due in no small part to advances in medicine. Death is a normal process for humans. The cause of death is the irreparable malfunction of the human body at the end of the statistically expected life span. A hypothesis was formed that cancer is an evolutionary process that refreshes the genetic pool, a mechanism of adverse selection of mutant alleles [1], and it eliminates genome instability.

The conventional considerations assume that the increasing cell cycles in the growing number of cells (genes) [2] increases the probability of malignancy, which increases according to the number of cells in the system. However, comparing cancer in various species contradicts this expectation, forming a paradox raised by Peto [3,4]. Peto’s paradox is one of the intensively debated problems of comparative biology. The paradox has no solution to the observation that the incidence of cancer does not appear to correlate with the number of cells in an organism at a species level. The assumed principle that the risk is equal for each cell to become cancerous does not fit with cancer occurrence compared to humans, mice, or elephants. Due to the significant correlation (linear dependence [5]) of the incident and mortality ratios, the Peto paradox on cancer incidence may be extended to mortality. The research interpreted the paradox as not the number of cells connected to the prevalence of malignancies but the study of the pool of stem cells and the specific microenvironment of the cells, which may help resolve the paradox [6,7]. However, this supports the Peto paradox that more tissue stem cell divisions have a higher cancer risk in humans [8,9].

Another of Peto’s “paradoxical” observations started with an in vitro study that repeatedly treated mice with a mutagen for many months [10]. He observed that the cancer incidence rate increased as a power function of the duration of exposure to the carcinogen, with a 4–6 power value, which appeared to be independent of the body size of mice. Monitoring general data collection for humans across different countries supported the power law [11]. The medical activity at the unexpected end of life focuses on age-dependent diseases to expand the expected life span of humans. However, the growing age of humans rapidly increases cancer incidence and mortality (CIM). Gerontologic diseases (also like cancer) rise due to the loss of the specific functions of the parts of the system. The CIM behaves according to a calling power law:(1)$$CIM\propto {age}^{\vartheta }$$
where ϑ≈6–7 according to many observations [12,13]. Another statistical evaluation of the age-specific incidence of skin, colon, and pancreas cancers with the age of Canadian men in 1970–1972 shows exponential growth by age with 4–7 exponents, which is a straight line in double logarithmic scales [14].

The somatic mutation theory of carcinogenesis claims that not just one but several mutations are probably required [15]. The successive mutations in the same cell initiate cancer with ϑ≈7 [11,16]. This epidemiology observation allows us to describe the cancer prevalence via the independent probabilities of six subsequent mutational steps, having essential complexity [17], breaking with the early view that the tumor process arises from developing a single renegade cell [18]. The renegade-cell concept purports that the malignant tumor originates from a single cell, which goes through many stages before producing a clinically noticeable condition [19]. This concept may be harmonized with the multistep subsequent series of changes, which, in the end, manifested as a malignant tumor. This localized explanation excludes the changes in the cellular environment on the way to becoming malignant. Another model describes the power law with two independent (nonlocal) time parameters as the first manifestation of the malignant cell and the growth time of the detectable tumor, supposing their normal (Gaussian) distribution [20].

The intensive research in cancer biochemistry turns attention to the molecular processes instead of the intercellular and intertissue influences, which apparently support the internal driving force of cancer development [21]. This research supports the random occurrence of malignant processes. Cancer development debates focus on whether environmental or random (“bad luck”) processes have a decisional role. The “bad luck” hypothesis is supported by the significant occurrence of cancer (~70%) from random errors throughout DNA replication in healthy stem cells, an unpreventable internal malignancy cause [22]. However, energetically open living systems cannot separate from their environment, and the synergy of the extrinsic and intrinsic factors is considered in cancer etiology [9]. The strong correlation between tissue-specific cancer risk and the lifetime number of tissue-specific stem-cell divisions does not mean that the internal processes happen independently from external influence. There is evidence that intrinsic risk factors contribute only modestly (less than ~10–30% of lifetime risk) to cancer development, so the inherent effects are heavily influenced by environmental changes [23].

According to our current knowledge, there can be many reasons for developing a malignant tumor. Some of these are biological characteristics depending on the condition, others require medical intervention, and a large group of causes can be treated individually, controlling the lifestyle. Figure 1 shows some of the most known causes of cancer. Naturally, we have no idea why those 4–7 factors accumulate and cause the clinical symptoms.

Physicians commonly note that “Any drug with no side effects has no intended effect either”. This simple observation summarizes the problem of living complexity. When our intention acts into the desired process, we influence related activities of the body’s physiology, where the chosen effect is complexly embedded. The multiple-interacted, self-regulatory system is complex [24]. All actions in the homeostatic equilibrium induce compensatory processes and make a positive selection to keep the defense mechanisms. In this way, any drug therapies, together with their targeted effect, cause an opposite compensatory process, developing resistance against the impact of the drug. This simple rule is similar to the Le Chatelier–Braun principle in chemistry and fits the Darwinian selection principles in general through the development of species. Nowadays, an example is antibiotic resistance for a significant number of the population due to the intensive antibiotic therapeutic and environmental load.

Complexity does not mean complication but the intertwining of processes which, at each step, seeks to have a dynamic and interconnected balance. Various regulatory mechanisms maintain balance or homeostasis. These mechanisms often involve pairs of opposing regulators to ensure that a specific biological process stays within a certain range or set point by using generalized suppressor–promoter pairs of the regulatory function [25].

The living system’s characteristic stochastic (probability) behavior is related to the intrinsic bifurcation by promoter–suppressor balancing. The simplest bifurcation model is a nonlinear double-well potential of chemical reactions [26,27], and, in general, it aims for suppressor–promoter balance. (It works like tossing a coin to occupy one of the potential wells in the bifurcation scheme.) A simple bifurcation in the influence of promoter–suppressor influences could help us to understand the “edge of the chaos” phenomenon of living organisms [28,29]. The life processes must keep their dynamic, energized form. The decisional macro-driver is homeostasis, which controls global physiological regulations. This systemic process is tightly connected to the internal and external connections and drives the actual feedback to regulate the relationships between the nodes in the microscopic range.

When their energy at the energy breaking point is too low, the process stops and “freezes” in one of the potential wells. However, when the provided energy is too high, the system loses control, and the promoter–suppressor balance cannot regulate the processes. As Einstein formulated, “Life is riding a bicycle. To keep your balance, you must keep moving” [30]. A characteristic example is the catabolism of humans, where the dynamically acting electrons drive the positive feedback: “Life is nothing but an electron looking for a place to rest” [31]. The common idea that biosystems evolve toward a stable equilibrium is a misperception of reality. The balance is only at the level of the dynamic (time-dependent) processes in the general homeostasis.

Predominantly negative feedback characterizes the regulation mechanisms. Homeostasis is often ignored and used as a static framework for effects [32]. The challenge is the complexity of living organs and the highly interconnected interactions between the parts. The demand to understand and use dynamic equilibrium develops a new paradigm for investigating living matter, which requires a stochastic approach (probability of events dependent on time) instead of conventional thinking that requires deterministic changes [33]. The dynamic homeostatic equilibrium keeps the system stable but constantly changing.

The homeostatic functions characterize the local stability of the living system, having very complex feedback mechanisms that secure the strength against a relatively wide range of perturbations. The homeostasis is not static. It is a self-organized dynamic process. The system is energetically open when frozen to a static state that is death. The complexity of the dynamic behavior guarantees robust stability, so the system is in a homeodynamic position rather than a homeostatic one.

The living system exchanges energy and cross-transports materials and information with its environment. Like the cells, tissues, and organs, every body part has open energy trade with other system parts. The spatiotemporal arrangements of the living organisms and their components are self-organized [34,35]. Self-organization explains the system’s evolution [36], expressed in nonlinear dynamics [34]. The self-organized feedback secures stability against a relatively wide range of perturbations. The structures’ self-similar building simplifies their construction by deterministically or statistically repeating the same template and connecting them with the same network [37], building a self-similar harmony. The self-organizing develops the fractal structures [38] of the biological objects, characterizing the pathological clustering of the cells [39].

The complexity of the dynamic interaction represents a feedback regulation of the system at every level of its structure. The complex system cannot be considered a sum of its distinct parts. The whole is more than the sum of the elements, the interactions are primarily nonlinear, and the system is energetically open and has adaptive exchanges with its environment. The approach to describing it must be analytic and not synthetic. Considerations regarding the complexity create considerable challenges in making the calculations. The attempted solution typically synthesizes the parts that could be calculated. However, this calculation strategy needs to be revised. The analysis must consider the complexity.

## 2. Allometric Scaling and Peto’s Paradox

Using the self-similarity and self-organized behavior of the living objects, a generalized comparison could be developed by the allometry of the species. Allometry is the size relations of an organism and any of its parts, using the scaling behavior of the self-organized systems between the metabolism and mass of the species [40]. It could be generalized for many biological functions in their mass dependence [41]. The energy transport to metabolic processes flows through the cell membrane. The surface-driven nutrient feeds the cell volume. Assumed to have R, the radius of the group of fed cells, the cells’ surface ratio to its volume characterizes the ratio of the basal metabolic rate ($BMR\sim R^{2}$) and the mass ($M\sim R^{3}$): ${BMR}^{1/2}\sim M^{1/3}$. Consequently, the scaling of the interspecies allometric relation should be $BMR\sim M^{2/3}$. However, the reality differs, and the observed relation is $BMR\sim M^{3/4}$. Hence, the observed BMR pro unit mass is
(2)$$B_{0}=\frac{BMR}{M}=M^{\beta }\sim M^{-1/4}$$

The allometry comparing various living species resolves Peto’s paradox of cell density in cancer prevalence. The BMR in unit mass has a scale-free power law (2). The species’ mass and the involved cell number are proportionally related [42], as well as the metabolic rate proportional to the resource delivery [43]. Using these results, the waiting time for cancer occurrence has a scaling exponent of 0.22 [44]. On the other hand, we know that the life span of mammals has a scale exponent of 0.21 [45], and β=0.20 [46], which are very close to the cancer-occurrence time scaling. The time to the occurrence of cancer in a subject’s life varies with the subject’s average noncancerous end of life. Due to the delivery rate of resources, $B_{0}$ decreases by scaling with β=−0.25 (2). At the same time, the mammalian life span and the waiting time for cancer occurrence grow by β=+0.25 scaling [47]. Lifetime energy expenditure has a very slight scaling dependence (β=−0.09) on the body mass, as measured for mammalian species [48]. The Peto “paradox” is solved this way, equalizing the life span increase by the decreased metabolic rate in unit mass. Indeed, the heartbeat/life span is approximately the same in all mammals, independent of their mass or life expectancy [49]. Corresponding to the scaling behavior, mice collect mutations at a higher rate than the organisms with more cells (larger mass), so they could have cancer in a much shorter period [50] despite their relatively low cell number. Allometric scaling is an efficient resource allocation connected to optimal energy utilization and minimal entropy production, often connected to the homeostatic regulation to maintain the internal stability of the organism despite the different body sizes. Growing body mass decreases the cells’ energy expenditure on average and optimizes the collective, cooperative functions of the body.

Cancer prevalence and growth follow ecological stoichiometry, which refers to the balance of different chemical elements within living organisms, particularly the ratios of essential elements like carbon, nitrogen, phosphorus, and other nutrients [51]. The ecological stoichiometry appears as a further modification, which may be combined with the allometric scaling supporting the solution of Peto’s paradox [52]. However, the paradox of cancer incidence and mortality needs deeper insight into the biological structure and dynamics. Allometric considerations can be used in intraspecies and intraorganisms to study cancer development. The self-similar self-organizing process is collective [53] and relates to the allometric scaling of living species [54,55]. Allometric growth of tumors has been proven mathematically and via in vivo experiments [56]. Allometric scaling is valid for the cancer and its subsystems [57]. Allometry gives the possibility to describe the development of the tumor [58]. It is helpful for primary cancer lesions, but it is not always applicable in metastases [59].

Healthy homeostasis regulates the system and optimizes energy expenditure, minimizing entropy production and maximizing the efficacy of the available energy. Entropy refers to the degree of disorder or randomness in the living systems, counting to the loss of organization and energy efficiency during biological processes. The first formulation of the minimum entropy production for the noncontinuous system was performed by Prigogine [60,61]. This theorem was generalized by introducing the order of stationarity [62], dealing with the dynamics and stability of the stationer systems [63,64]. Normal cellular division inherently involves some entropy production in a balanced energy flow and cellular organization. Cancer cells exhibit higher entropy production compared to normal cells. This is due to uncontrolled proliferation, leading to increased energy expenditure and disorganized growth patterns. The metabolic alterations of cancer cells often rely on inefficient metabolic pathways for energy production, generating more heat and waste products. Furthermore, genetic mutations can disrupt protein function and cellular regulation, contributing to disordered energy flow and increased entropy production. The specific energy usage ($B_{0}$ in (2)) indicates that the energy efficacy grows with the mass of the organism, having decreased energy utilization in the unit mass. This better energy usage may limit the increased cellular heat production as the tumor thermogenesis, which is accompanied by increased entropy. The limited thermogenesis ultimately limits the higher entropy production and could be involved in the cancer prevalence as a further addition to understanding Peto’s paradox.

## 3. The Power Law of Cancer Prevalence

The homeostatic control and the minimal entropy production optimize the energy utilization, which could have a pivotal role in cancer prevalence. Random genetic mutations offer a linear expectation of cancer incidence, but (again paradoxically) they have a power-law function, indicating that a number of sequencing mutations have a decisional role in cancer causes. The observed epidemiologic power function (1) means, practically, that a person collects a definite number of mutations until manifesting cancer symptoms [12,13,14]. A total of 4–7 mutations in sequential series cause cancer [60,65]. The epidemiologic observation of cancer incidence involves naïve individuals who were not treated with antitumor therapy until the proven clinical manifestation of cancer. The further development of the cancer history depends on the cancer therapy.

Homeostatic control deals with environmental changes and inherited and spontaneous random mutations of the genetic network. DNA replication is not perfect; it modifies individual sequences of the genome. During DNA replication, random mutations are inevitable. Environmental stress and complex interactions induce point mutations, and their adaptive amplification was thought to promote genetic changes to enhance survival [61,66]. This interpretation does not describe the complete situation. A random mutation creates unfavorable results because there are considerably more ways to damage than improve the genome. Genes autocatalytically and individually (“selfish” way [62,67]) synthesize their sequences. Random mutations are challenging: they are highly likely to be disadvantageous, and adaptive benefits are less likely to materialize [63,68].

Due to the compensatory dynamics (reversion), all stresses develop their reverse reactions, and even cancer therapies may induce malignancy by reversing mechanisms [64,69]. The selection pressure may favor mutants resistant to the external cytotoxic effects in the subsequent cellular generations [65,70]. Some random and compensatory mutations present better fitness for survival to adapt to the larger scale of internal and external challenges. These well-adapted cells become the driving force of evolution over a very long timescale.

The change is always evaluated based on the cost/benefit ratio in a short decision-making time frame. Statistically, random mutation is an unfavorable event. Thus, a dilemma appears: to “invest” (cost) in correcting the error or to leave it because it is marginal. The cost to repair is more than the benefit of leaving it unrepaired. If the mutation is potentially lethal, the benefits of action outweigh the costs, and reparations are mandatory. The repair is expensive but avoids the worst, so the payoff is higher than the cost. After restoration, homeostatic regulation resumes, more random mutations occur, and the cycle repeats. The situation is different when the mutation is not lethal, the damage is marginal, and the benefit of the action is small compared to the repair cost. In this case, the damage remains uncorrected [63,68]. Over time, more random mutations appear. Even though the uncorrected mutations are individually marginal, they, together in a set, can pose a severe challenge in maintaining homeostasis. The cost of reparation of accumulated mutations increases; thus, the probability of reparation decreases. The collection of mutations presents multifaceted challenges, and the increasing costs may be too high to repair. The system tries to find other solutions to survive. The adaptation answer on the induced genetic or epigenetic changes increases cancer’s probability (Figure 2).

Cancer paralyzes the organism, using its resources, contradicting the systemic cooperative demands. The cancerous development has “endless complexity” [17]. There are various theories and hypotheses about the cause and origin of cancer, from ancient medicine to many new explanations. The virus concept developed over a century ago was one of the first modern explanations [66,67,71,72]. Later, the genetic clues were favored [68,69,73,74], and the mutation concepts became widespread [11,70,71,75,76]. Recently, the immune dependences [72,77] and connections with wound repair have been intensively researched [73,74,75,76,77,78,79,80,81,82]. Despite the enormous efforts, the cause of cancer remains open [78,83]. Due to the commonly accepted individually manageable sources of cancerous diseases (see Figure 1), studies turn to the environmental, diet, and habit origins of malignant diseases [79,80,81,84,85,86]. No general explanation for the cause of cancer has been developed, and the intensive search for answers remains challenging [82,87]. Despite even particular quantum-physical reasons [83,88], the recent studies do not give a final solution [84,85,89,90]. The epidemiologic power law (1), as a statistical proof [12,13,14], may offer a clue for an explanation of the cancerous process.

The standard model of carcinogenesis usually applies a linear configuration of the development. The cancer development contradicts the oversimplified reductionist deterministic approach. Complex tumor development breaks the early view that tumors arise from a single renegade cell [18]. The explanation of the process cannot be linear. The Darwinian selection model needs to consider nonlinear dynamics, which studies the cellular genetic instability in the frame of the competition of genetic strategies. Cancer is a stochastic process with numerous complex and interacting hallmarks in its development [86,87,91,92]. The healthy host supports cancer development [88,93], while the growth of cancer dismantles the multicellularity [89,94], and the cellular collectivity gradually disappears [90,95]. This development is similar to atavism [91,96], but the active support from the host tissue distinguishes cancer from the passive availability of resources from the environment in unicellular (early evolution) conditions.

Darwinian selection is based on random changes but has a definite driving force: the cost/benefit balance, which accounts for complex internal and external conditions. Balancing requires a decision process that determines natural selection within and between species. Natural selection is active in all groupings of living systems, from group selection of species through the individuals to the genetic level of preference. The random mutation process optimizes the cost/benefit balance. The decision-making strategy could be individual when the competitive character is emphasized and could be collective when the benefits of cooperation dominate [92,97].

The evolutional game theory could successfully model cellular decision making [93,98]. The game style of the evolving tumor changes the cooperative healthy stromatic collectivism to competitive individual decisions answering on the random challenges. Cancer develops a competitive cellular game, but the principal driving force remains: the cost/benefit ratio drives independently from the other cells. The cellular individualism of cancer cells replaces the cooperative multicellularity. The individual cancer cells compete with surrounding cells for resources. The genetic network evolves from generation to generation as a “clock” measuring the nonmonotonic self-time of cancer development.

The tumor develops genetically in subsequent cell populations, not in a single cell. The Darwinian evolution for species is valid for the selective development of the cells during the line of successive cellular divisions when the cell fitness decides its viability. Cellular selection is based on the cell’s genetic information, so it is not the cell. Still, it is molecular information that is selected during a process, and the cell is only a carrier of the genetic structure. The selection of the randomly mutated genes (mutated cells) is already present in the population of multicellular organisms. Their fate can be interpreted based on evolution, similar to the population level of species, which is guided by Darwin’s law. The most viable cell wins. The nonlinear Darwinian considerations may divide the cancer development into six distinguishable steps when the game strategy varies due to the changes in the challenges under the permanently created random mutations (Figure 3).

### 3.1. First Stage of Cancer Development

The cells are not protected against environmental mutagenic attacks. The mutated single gene rarely causes disease (also no cancer). The perturbations triggering disease affect the complex intracellular network and, consequently, the extracellular network.

The cancer cells have extended adaptability to external conditions. The primary step of cancer development evaluates the mutational damage marginal and benefits from the costless flight, as shown in Figure 2. This ignorance develops through numerous cell divisions, repeating and extending the mutational damages. Some mutations are inherited and indicate a risk of cancer connected to the mutated genes [94,99], and some develop de novo and are inherited in the next cellular generation. The cancer cells also may have epigenetic mutations [95,100]. The cellular division replicates some epigenetic information with DNA methylation [96,101].

Resisting and correcting errors takes time and energy, while ignoring those attacks, which are not lethal, has no cost. Consequently, errors (mutations) may be collected. The process depends on the frequency of the episodes, so the development is time-dependent. The accumulating mutagenic defects need increasing energy to repair, so ignoring those has an advantage when they do not affect the cell’s vitality [97,102]. The genetic instability in mutagenic environments develops because DNA repair requests too much energy [63,68]. The behind DNA repair in normal, nonmutagenic conditions also has an unbalanced payoff–cost game. The cost of DNA repair is primarily independent of the genetic location, but the cost of ignoring the error depends on its sensitivity; a slight change in the nucleotide chain has a wide range of minor-to-severe consequences [63,68]. The balance of cost/payoff preserving genetic stability is asymmetric and depends on the error’s type, occurrence frequency, and replication rate. The chromosomal instability initiates the carcinogenesis [98,103]. In this way, carcinogenesis is based on molecular evolution in the Darwinian nonlinear way. It unites the genetic and environmental influences in cancer development [97,102]. Carcinogenesis is based on DNA mutations. The uncorrected collection of random mutations has increased the repair cost, so its growth is undisturbed, developing other survival strategies (Figure 2).

The collection of random, individually marginal mutations could be dangerous. The collected mutations transform the cells seeking selfish individual survival. It breaks the multicellular cooperation, ignoring the cooperative common interest. Malignant cells develop competition for their discrete survival. The accumulation of the mutations depends on the tissue type and the scale of random mutations. The cell represents a high particular vitality, and the surrounding stromal network ignores (or even supports) uncooperative behavior. Supporting uncooperative individual cells by the cooperative stromal network could be expected because the normal cellular division to deliver new cells is a standard way of healthy development. The surrounding stromal host does not recognize the malignancy at the beginning. The cell looks “normal” and only has a collection of ignored (individually not dangerous) mutations.

While genetic instability is expected to limit the cellular divisions [99,100,104,105], the cancer cells, contrary to their genetic instability, have faster growth rates than healthy cells. The relatively stable complex arrangement uses dynamic equilibria. It balances by “decisional” bifurcations of the participating molecules that “decide” their behavior: it would be representing a “friend” or “foe”. The usual promoter–suppressor balance appears at the molecular level as the “Janus face” of the functioning of critical molecules. It seeks to answer what they are: enemies or friends in a collection of marginal and unattended mutations. The challenge of the “double-edged sword” behavior is typical for complex regulations, balancing multiple oppositional regulatory feedback and determining a window of decisions. Complete molecular groups, which have an essential role in diseases [101,106], also behave in a frustrating way for cancer development: it could have an antitumor action, causing apoptosis, or it could be a promoter of the tumor progression [102,107]. Such protein families (chaperoning stress proteins, SPs [103,108]) have a critical role in establishing weak links for stable, nonvulnerable networks [104,105,109,110]. SPs have typical double-edged sword phenomena [106,107,111,112], including the antibodies against them [108,113]. The conditions determine the position of “friend or foe” [108,109,110,113,114,115], according to the Darwinian selection principle. The relatively stable complex arrangement uses equilibria that “decide” the division of the participating molecules, and their behavior represents “friend or foe”. The friend or foe distinction is rather complex in malignancy, where the cancer cells function correctly in their individual life, having solid and vivid immortal behavior; however, their activity is destructive to the system they are a part of.

The unicellular malignant proliferation and protozoan-like invasion as a stage of evolution share many standard features [111,116]. In this comparison, cancer is an atavism [96], judging by its self-regulated unicellular growth. However, this similarity has its limits. The loss of multicellularity is an atavism, indeed. Still, the intense onslaught of the healthy network that forces it to care for the tumor growth differs from the unicellular state at earlier stages of evolution.

The malignant cells do not have the benefits of collectivism, but there is no cost to join the network. The cost/payoff ratio is <1, so cellular individualism is beneficial and has become dominant [112,117]. The autonomic conditions of cancer cells become profitable only as long as the growth enjoys nutrient-rich environmental conditions for survival [113,118]. However, when food becomes insufficient and healthy homeostatic regulation attacks single-cell growth in constricting conditions, cancer cells develop a solution to survive, tipping the balance in favor of “friend” or “foe”. The cancer process avoids natural apoptosis [35,114,119]. The SPs protect the malignant cells and appear as a “foe” of the organism in these processes.

The game theory offers a proper decision-making method in all frustrated situations, counting the cost/benefit ratio among the random mutation-making conditions, thus optimizing the game by using a mixed strategy. Darwinian selection prevails, guided by the decisions for optimal adaptation to the circumstances.

Collecting six cancer development mutations to manifest in the clinical stage is incorrect. Multiple mutations are necessary for the first stage to produce precancerous cells. The occasional mutation of particular genes continues the first-stage process, and, at the end, we may recognize approximately six stages by the time the cancer is clinically manifested. All stages have many mutations.

The formation of the first stage of cancer is a long process. A considerable amount of cellular division is necessary to find the adaptation mechanism. Unsuitable cells are negatively selected during the process, while cells that can adapt to harsh conditions survive. The adapted cells are precancerous and can continue their unicellular life. What is beneficial on the level of the cells may be deteriorative, deleterious, or harmful on the system level. The individual precancerous cells are more viable than their stromal neighbors, ignoring the collective demands (not paying the cost of collectivism) and being tolerated by its healthy environment. It is a reasonable basis for the next stage of cancer development.

### 3.2. Second Stage of Cancer Development

Occasionally, a random mutation may modify a gatekeeper gene. The caretaker genes’ mutation (or epimutation) follows the first step during the random genetic changes. The caretaker genes prevent mutations from being inherited in the next cell cycle by triggering apoptosis or slowing the cell division, allowing the proper DNA reparation, so its effect is indirect in the regulation of cell proliferation [115,120]. More mutations added to the gatekeeper gene in the same cell induce cellular breakdown with a high probability. The caretaker gene controls the integrity of the genome, and its mutation could cause genetic instability without cellular breakdown. This gene does not directly act on cellular proliferation but regulates the cell-cycle checkpoints, DNA repair, and apoptosis. Consequently, this gene maintains the overall genetic stability [116,121]. The genomic instability could be developed by changes in the nucleotide sequence of DNA or by the faulty arrangement of chromosomes [117,122]. The random errors in the DNA replication appear as a risk of malignancy. The caretaker functions to fix the proper sequences, preventing the accumulation of mutations in the genome.

The caretaker gene controls the correct cell-cycle checkpoints. The checkpoints have special functions in the cellular division as G1 checks for the cell size, growth factors, and DNA damage; G2 checks for DNA replication, and the spindle assembly checkpoint is responsible for chromosome attachment to the spindle. The caretaker gene controls all of these. The DNA repair mechanisms ensure genome stabilization, so the mutation on the caretaker leads to uncontrolled cell proliferation, which could cause cellular immortality and cancer. The caretaker tumor suppressor gene mutation allows for cell division more often than usual and does not force apoptosis for the cells with damaged DNA, so it unleashes improper proliferation [118,123]. The accumulation of mutated cells depends on the cluster size and tissue architecture of stem and differentiated cell compartments, and the local regulation of homeostasis favors chromosomal instability [119,124]. The caretaker genes are supposed to be involved in many hereditary cancer tendencies [120,125].

The unleashed cell evolves into an individual fight-or-flight conflict game, which is commonly used in biology [121,126] and evolutionary strategies [122,127]. The conflict in such a strategy is usually named a Hawk–Dove game [123,128]. The population has aggressive (Hawk) cancer cells and collective (Dove) stromal cells. The fitness of cells determines their aggressiveness [124,129]. The cooperating stromal cells share their payoff, while the determinedly fighting cancer cells risk considerable loss by competitively fighting with other cancer cells. Nevertheless, when a cancer cell wins, it has a high payoff (while the defeated cancer cell has a high cost and dies). When the aggressive cancer cell meets the stromal compartment, the last flight has no payoff, and the cancer cell wins but gets less of what it would have gained by defeating another cancer cell in a fight. The situation forms a Nash equilibrium when the cost of change forces one part of the process to balance the benefit that could be reached; no further actions would produce more payoff than the cost for both parties; it is a win-win position [125,130]. When the cancer cells dominate the population, the fight between them will be frequent, and the loss, on average, will be high in cost and low in terms of its compensating benefit. If the stromal cells dominate, they are an “easy target” for the cancer cells obtaining payoff, while the stromal cell has no benefit on average. In the dominance of stromal cells, the cancer cells have considerable payoff from the frequent meeting with the stromal environment, so their population grows again, reaching a situation where competition with the other cancer cells is more costly than beneficial. In this way, it forms a Nash equilibrium, which depends on the costs and payoffs of the fights and flights. This equilibrium remains as long as the cost/benefit ratio does not change and equilibrium exists.

There is no pure strategy in these games. It is relatively easy to defeat a pure strategy by developing a counteraction and finding the antidote primarily by immune monitoring. The pure strategy is unstable, and, consequently, the uniformly responding population is unstable. Stable cancer growth requires extensive heterogeneity. For stability, the cancer has a heterogeneous structure both genotypically and phenotypically [126,131]. Cancer development uses a mixed strategy via random mutations in the natural processes. The hybrid strategy forms an evolutionary stable solution (ESS) [127,132].

The heterogenic structure of consequence of Darwinian evolution’s randomly mixed (hybrid) game strategy builds different TMEs, allowing for different cost/benefit ratios in the local volume. These alterations form variants of the ESS Nash equilibrium [128,133], allowing remarkable regional differences. The cancer heterogeneity varies the payoffs [129,134] and breaks the uniform game. When the Nash equilibrium favors a larger stromal cell population in the local volume, it selects those cancer cells from the heterogenic pool which are more aggressive. It shifts the Nash equilibrium to their advantage. However, heterogeneity contradicts Darwinian evolution, which preserves the most robust, viable variants that best adapt to a given condition. This paradoxical fact also appears in the evolution of natural selection between species, where the vast variation and diversity within the same species seem to contradict the selection of the most robust and most adaptable breeds, replacing the weaker types. This happens in the multiparticipant competitive game. A strong individual primarily fights with the other strong individual when the possible win gives a greater payoff, considering the strategic benefit of fitness maximization. These selected fights leave the weaker bread relatively unchallenged in the probability of their survival growth. The weakest has the highest fitness [130,135]. In consequence, the more fragile type also stays in the population. This particular selection gives a chance for the weak type of genetic mutations to survive in the competitive game for further multiplications.

### 3.3. Third Stage of Cancer Development

A random target shot damages the gatekeeper genes in the third step. The gatekeeper genes encode a system of checks and balances that monitor cell division, and death may arrest the potentially pathologic processes in the cell division [131,136]. The observation of the mutation of the gatekeeper gene in colorectal tumors was the origin of the naming of this tumor suppressor gene [132,137]. Gatekeeper genes regulate the functional growth rate, balancing between the proto-oncogenes and tumor suppressor genes [120,125]. Its damage can unleash the unicellular growth, and the proliferation becomes uncontrolled. However, when the gatekeeper gene is functional, the imbalance of the other genes is inhibited, but irregular growth and differentiation happen when this gene is mutated [119,124]. Only a few gatekeeper genes are present in each cell, but these are tissue-specific [131,136], carry the mutation inherited from the progenitor, and persist during cell replication [119,124]. The gatekeeper gene mutation increases the probability of the mutations of other genes in the DNA replication mechanisms. Gatekeeper genes regulate apoptosis, preventing enhanced cellular divisions in normal homeostatic regulation. The gatekeeper mutation may inactivate this regulating mechanism, opening the way to uncontrolled cellular proliferation. The cell becomes independent and loses most networking connections, creating unicellular development. The cancer risk due to affected gatekeeper gene populations is much more than with caretaker genes [120,125].

When the blood supply cannot deliver enough oxygen, many tumor cells switch to oxygen-independent glycolysis (Warburg effect [133,138]). The immediate massive demand to energize the mutated cellular processes rearranges the intracellular transport. Levy’s flight [134,139] describes the complex heterogeneity of the diffusion in the cytosol, needing support from the cytoskeleton [135,136,140,141]. In proliferative conditions, the cytoskeleton partly or completely collapses [137,142]. This limits the transports to and from the mitochondria and favors the simple cytosolic, nonmitochondrial anaerobic glycolysis. Anaerobic glycolysis produces rapid ATP production but has lower efficacy than mitochondrial phosphorylation [138,143]. The change from standard phosphorylation to fermentative metabolism is one of the factors causing cellular autonomy. The anaerobic glycolytic phenotype likely induces cell invasiveness [139,144]. The change in metabolic pathways is essential for malignant development. 

The massive ATP production increases the liberated heat, generating a temperature gradient between the extra- and intracellular electrolytes. The conductive heat flow turns to convective [140,145], causing Bernard instability [141,146]. The convective heat flow stimulates ionic transport through the cellular membrane, which increases glucose intake. This supports fermentative metabolism and changes intracellular circulations [142,143,147,148], downregulating mitochondrial phosphorylation metabolism.

Tumors have extended metabolic heterogeneity. The adaptation mechanisms of cancer cells develop different strategies depending on the cancer heterogeneity. One is the reverse Warburg effect [144,145,149,150]. The bone marrow-produced mesenchymal stromal cells may interact intensively with cancer cells. The glycolytic pyruvate → lactate transformation direction may turn oppositely, and the oxidative cell supports the lactate balance [146,151]. The hypoxic stromal cells have a metabolic symbiosis with oxidative tumor cells. The cancer cells trigger reactive oxidative species (ROS) in hypoxic stroma cells, and the oxidative cell converts lactate to pyruvate. The consequent oxidative phosphorylation processes in mitochondria in cancer cells produce ROS to hypoxic stroma cells, which has a positively accelerated feedback loop [146,151]. The reverse Warburg effect needs an adequate oxygen supply [147,152], which is unavailable in all tumorous cases and depends on the disease stage. This metabolic coupling [145,150] shifts the research from the oxygen supply, a single tumor cell-associated Warburg effect, to the molecular interactions of cancer and stromal cancer cells with the neighboring cancer-associated fibroblasts [148,153]. The Warburg effect and its reverse process changes the game. Due to the hypoxic environment being poisonous for the healthy somatic cells, the game between the tumor and its host turns into a prisoner’s dilemma [149,154]: cooperation is an optimal bet for normal cells to strengthen their competitive abilities (metabolic benefits), and the cancer cells continue their hypoxic production. In this situation, neither party is interested in breaking the balance; that is, they create a Nash equilibrium. In this way, the cancer cells metabolically cooperate with adjacent supporting stromal cells. The cooperation is based on exchanging different catabolites, like monocarboxylate lactate, pyruvate, and ketone bodies [145,150]. The metabolic coupling could be used for new therapy concepts and prognostic biomarkers and is also an excellent predictive possibility [144,149].

### 3.4. Fourth Stage of Cancer Development

This stage starts with the mutation of the landscaper genes, which do not directly impact cellular growth. Still, their mutation stimulates the neighboring stromal cells to support the proliferation [150,155]. The landscaper genes control the microenvironment of the extracellular matrix (ECM, developing TME) by cellular membrane proteins, molecules of adherent connections, growth factors, etc. [132,137]. It promotes cellular communication via direct contact and signaling transmitter molecules [151,156]. The stromal cells transformed to precancerous cells, inducing abnormalities via the mutated landscaper genes [152,157]. The membrane changes could support abnormal proliferation and differentiation, and the favorable environment of the ECM could modify the immune-cell activity in the volume [151,156].

The interactions between tumor cells (including stromal and malignant cells) are not always exclusively competitive. Tumor cells with different capabilities cooperate with stromal cells to obtain the necessary nutrients for survival. At the start of malignancy, the situation probably corresponds with the infinite availability of nutrition for the precancerous cell. Still, the growing malignant population needs more nutrients for further proliferation. The cancer hallmarks do not characterize single cancer cells, which are very diverse. The hallmarks describe the group of cells that collectively has them. A new game strategy appears whereby the tumor-connected stromal cells not only “flight” when cancer cell attacks but also actively participate in tumor growth, having a primary role in the processes in TME [153,158] and/or using the reverse Warburg effect [145,150].

Binary cooperation between the tumor and stromal cells starts with the mutated landscaper genes, which prepare the game change when the resources are limited and could inhibit the proliferation. Cancer cells start cooperating to survive and create a cooperative game instead of competing, ensuring the demand for proliferative nutrients. The cluster of tumor cells adapts to the host like in the Darwinian evolution of species. The individual typically adapts to the environment to survive.

The fourth step changes the individual competitive (noncooperative) game of cancer cells to the cooperatively working malignancy. The malignant cells (as players) can achieve superior outcomes by working together rather than against each other. Cooperative play assumes that players come together to compete for the highest payoff level, considering the external enforcement. The heavily mutated cells force the mutualism with the reverse Warburg effect and build up a complex system that also evades the immune surveillance, the actual “predator” group for the unicellular structures. In general, “predator avoidance” is a clustering factor, such as weak multicellular organization [154,159]. The dense structure and expected hypoxic behavior in a large group of malignant cells protect against immune attacks. In addition to the defense, the rigid ECM [155,160] and the expanded pressure [156,161] promote the malignant invasion.

The decisional rationality of selfish players changes by the population dynamics of the cooperative game driven by Darwinian fitness and ensuring stability [157,162]. The reorganization for a cooperative game needs to “sacrifice” a part of the individual energy budget in exchange for the possibility of joint survival. The driving force of the limited resources to the direction of collectivity is observed in starving slime molds [158,159,163,164]. Collective actions help distribute the available, limited energy intake to share between the parties based on their needs and, in this way, satisfy the group’s survival. Due to the metabolic resetting and the increasing intensity of the Warburg effect, the collective cancer became hypoxic, with the development of glycolysis instead of phosphorylated-ATP production, and it yielded lactate, which lowers the pH. The size of the colonies is also a matter [160,165]. The basal, resting metabolic rate in a unit of the cancer mass prefers the growing mass in suboptimal alimentation [161,166].

The mutations (or epimutation) in caretaker genes can also mutate gatekeeper genes [162,163,167,168]. The third and fourth steps to cancer prevalence are identical in this case. The development process is reduced to five steps only.

### 3.5. Fifth Stage of Cancer Development

Cancer causes changes in the central regulatory systems of the body: it changes the transport systems, inhibits immune regulation, modifies the hormonal balances, disseminates in the organism, and uses the resources of the healthy organism unilaterally.

Homeostatic control attempts to “cure” the cancerous disorder, handling it as a wound in the healthy host [73,78]. A charge transfer is taken from the healthy to the tumorous lesion to stimulate the cells to “heal the wound”. The current is powered by endogenous field strength, forming an “injury current” [164,169], which is an internal rebalancing of the charge distribution, driving wound healing [165,170]. The injury current ignites cellular migration [166,171] and supports the proliferation to heal the wound [167,168,172,173]. The electric field ignites the cell division, and the field strength correlates with its frequency of it [169,174]. The dynamic change in the space charge is a self-gaining positive feedback mechanism. The injury current induces cancer proliferation, which furthers the injury current. At the molecular level, the growth and repair genes (oncogenes and proto-oncogenes) have a central role in forming new tissues [170,171,175,176]. In the case of a wound, T cells, macrophages, and NK cells migrate to the injured tissue to remove the debris and the injured cells. The T cells, the monocytes, and the macrophages present both growth and repair factors [172,173,174,177,178,179]. The growth and repair genes have a bystander effect, which ignites the growth and repair factors there, too [175,180], and collects stem cells to the edge of the wound [176,181]. The wound healing finishes when the growth and repair genes lose their activity or when the tumor suppressor genes are activated [177,182], and homeostasis is then re-established. The wound healing remains permanent and never finishes in cancerous tumors [178,183]. Tumor growth initiates angiogenesis to achieve sufficient nutrition and promote dissemination [179,184]. This link was discovered early [180,185] as a “trick” for cancer cells to adapt to the conditions. 

Pathological conditions, such as tumor growth, can initiate vessel growth [33,179,184], which can follow allometry [181,186]. The suboptimal alimentation triggers the cancer cells to secrete some angiogenetic substances, promoting proliferation by angiogenesis [182,187]. Therefore, the tumor initiates vascularization [183,188]. Moreover, angiogenesis is a hallmark of tumor development [184,189], and its growth is angiogenesis-dependent [185,186,190,191]. The process is governed by the endothelial growth factor (EGF) and its specialized form, vascular endothelial growth factor (VEGF). Mature endothelial cells must allow for both acute and chronic vessel wall remodeling. The charge redistribution helps the angiogenesis to supply the tumorous growth [187,192].

Another systemic effect is that the cancer evades the immune system’s surveillance, thus disrupting the natural collective control throughout the body (Figure 4). The physiological homeostatic impact, including the metabolic resetting, creates a new balance until the cancer cells are in the subclinical phase. Still, the governing feedback turns positive with the clinical stage, and the growth of the tumor becomes exponential.

The organized population dynamics allow the tumor to act like an organ [188,193], robustly representing the expected demand of the malignant cells organized in the cancer to have the highest backing from their surroundings to survive [189,194]. The increased size of the cluster of cooperating cancer cells increases the complexity of the structure, proliferates more effectively, and can share resources. Cancer cells may affect the hormone system via cancer-derived mediators (biogenic amines, neurotransmitters, neurohormones, cytokines, immune mediators, etc.) and, with these, can stimulate the neuroendocrine centers resetting the homeostasis [190,195], setting the body to favor the cancer proliferation. Numerous changes create a new homeostasis, which helps the tumor grow (Figure 5).

### 3.6. Sixth Stage of Cancer Development

Cancer changes systemic, collective self-organizing and develops its selfish rules, remodeled structure, and altered transport system. The homeostatic physiological effects, including the metabolic resetting, create a new balance until the cancer cells are in the subclinical phase. Still, the governing feedback turns positive with the clinical stage. In this step of cancer development, the tumor is clinically manifested. The case became part of epidemiologic statistics, which does not register the subclinical appearance during the five previous stages. Some symptomatic cancer occurs in a significant portion (~40%) of the lifetime of total human [191,196]. Additional nonsymptomatic malignancy is observed in autopsies [192,197]. At ages 50+, probably everybody has some malignant neoplasms [193,198]. These observations prove that the malignant processes are age-dependent, and according to the epidemiological data, they are also environmentally determined [194,199]. Mutations and genetic instability can already be present at birth [195,200], so cancer is a natural consequence of aging [196,201].

The aging of the individual has three effects on the cancer development:General homeostatic regulation weakens due to aging [197,202]. This loss appears in the variability of the physiological characteristics of the systemic networks (e.g., blood circulation disorders, nerve degradation, deterioration of movement coordination, etc.). The reduced homeostatic capacity limits the healthy control of cell divisions and reduces the effectiveness of physiological reactions to disorders to hinder proliferation. On the other hand, however, aging reduces the rate of cell renewal. Less DNA replication means fewer random mutations, which suppress tumor growth.Aging is a statistical factor when the number of DNA replications increases the likelihood of mutations and the possibility of cancerous processes. The mutation sequences follow the five previous steps. Still, due to the random mutation and various games and decisions, the duration of the appearance of the clinical stage is not calculable. Naturally, it will be more likely by the elapsing time. Furthermore, aging downregulates the potent inhibitor of cell proliferation, the transforming growth factor beta (TGFβ), allowing malignant development [198,203]. So, age systemically adjusts the progression of cancer.Cancer and aging are, molecularly, two sides of the same coin [199,204]. The caretaker genes have a crucial balance between cancer and aging [200,205]. Its decreased activity increases the cancer risk, which may reduce the life span. In contrast, its increased activity may finally limit cancer development but decrease longevity by apoptosis and senescence. The tumor-suppressor mechanisms contribute to aging. The p53 induces cellular senescence while it is a tumor suppressor, so the aging and the tumor inhibition may negatively correlate [199,204]. On the other hand, the senescent cell may promote tumor progression with a bystander interaction [201,206]. Senescence may be a consequence of gatekeeper tumor suppressor mechanisms. Consequently, while the senescence is suppressed, the cancer probability may limit longevity due to the collected senescence cells [202,207].

Aging has a particular energy consumption condition. The number of heartbeats in the entire lifetime in mammals has surprising behavior: it does not change with life expectancy or with the body mass of the species having $7.3\pm 5.6\times 10^{8}$ heartbeat/lifetime [203,208]. This observation is the direct solution to Peto’s paradox. It looks like the energetics of all living cells of the mammal organisms predetermine the life span. The inverse relation of the allometric scale between the life span and heart rate of mammals marks its direct connection to the basal metabolic rate. According to this, the basal oxygen consumption of every living species is ~10 oxygen/atom/lifetime in the body ($\sim 10^{-8}$ $O_{2}$ molecule/heartbeat) [204,209]. It is an energetic balance of life without any paradoxical considerations. In this way, the heartbeat, as a transport process, is decisive in the death rate [49,203,208]. This energetic unity could be the possible reason for the “rivalry” [199] between cancer and aging.

The aging reduces weak-link connections [205,210], destabilizing the molecular networks and decreasing their resilience. The permanent random harmful processes caused by various stresses during aging make the healthy system [206,211]. Breaking weak links could separate the hub-centered subnetworks and divide the complete network into different equivalent classes. The random damage of weak links increases the noise of signals in the aging process, causing further damage [205,210]. There is a positive correlation between the cellular resistance to stress and the life span [207,212]. Consequently, the lowering of the resistance by aging is a positive feedback route. So, the resilience of the gene network, which controls the cellular stress response, is decisional in aging and longevity [207,212].

The cancerous process could be dormant, and the clinical manifestation could be postponed, or the cancer recurrence after treatments could appear a long time after the finished therapy. The dormant cells remain quiet until appropriate environmental circumstances start the proliferation again [208,213]. The cellular division is arrested in a dormant state, and the cell cycle blocks the G0→G1 phase [208,213]. Dormant cancer cells are clinically unobservable, with minimal residual disease or micrometastases remaining after an apparently successful primary tumor treatment [209,214].

The dormancy could have multiple reasons. It depends on angiogenesis [210,215] or immune-mediated or cellular dormancy depending on TME, hypoxic ECM, or endoplasmic reticulum stress [211,216]. Moreover, III collagen could develop dormancy around the quiet cell, and the malignancy “wakes up” only when the collagen level decreases [212,217]. However, the positive influence of dormancy, in which the tumor does not develop for a longer time, has a negative feature when the dormant cells survive the applied therapy and could be activated again as therapy-resistant development.

In the clinical state, the tumor grows exponentially and may be characterized by its doubling rate. The breast tumor doubling time had not changed in the past 80 y for breast cancer, despite the rapidly developed therapeutical modalities [213,218], showing the extreme adaptability of the malignant cells to toxic stress. Toxic exposure is more challenging; it may select the tolerant cells on this particular toxic stress, so these mutations have more facility to avoid apoptosis in otherwise collected random mutational states.

## 4. Discussion

The spatiotemporal arrangement of the living organisms and their parts are self-organized [34,219]. The structural principle of self-organization is self-similarity, which is a repetition of a template that is exactly or approximately similar to a part of itself. This makes a similar structure in magnification by some orders of magnitudes, allowing for a scaling by magnification. The magnified pattern could form a self-affine system, which has different amounts of scales in different space dimensions. The self-similar building of the structures simplifies their construction by deterministically or statistically repeating the same template and connecting them with the same structure. This type of self-similar harmony is dominantly common in the various “building blocks” of the living system as a whole. The healthy dynamism repeatedly correlates with metabolic circles and all other fundamental living processes. The emitted (measured) fluctuation components characterize the time-set of different interactions and energy exchanges, showing a correlation of the signal with its earlier value at time-lag, τ. The time delay informs about the similarity of the signal parts when the same microscopic change happens in the repeated molecular signal pathways. The time lag of the autocorrelation function carries information about the dynamism of the microstates. The autocorrelation shows the preferences of possible variants of the molecular reactions [220], the selection of their timing, and the ordering for the desired signal pathway or enzymatic actions. The homeostatic balance defines the autocorrelation of the set of signals of biological changes [221,222]. The previously understood and accepted basic evolutional dogma of unpredictable random mutations needs modification in light of the newest research [223]. The genetic transformations have some nonrandom controls. Some gene groups never form a genome in the presence of another particular gene group, and some genes strongly depend on the different presented gene families. There are rules for genetic mutation when genes may cooperate or have conflict with other present genes. In this way, until now, strictly random mutational structures have some possible predictions.

The living systems are open, dynamic structures, performing random stationary stochastic self-organizing processes [224]. The self-organizing procedure is defined by the spatial–temporal–fractal structure, which is self-similar both in space and time [225]. The deviation from the randomness is probably connected to the self-similarity of the living objects [226,227], which causes a scaling behavior. In this way, the scaling is a general behavior of the self-similar conditions.

The fundamental phenomenon behind the scaling is the proportional relative changes in the parameters [228]. The living organism has to grow in collective harmony, so the relative growth of parts must be proportional growth [229]. Consequently, when the parameter ℘ depends on the parameter g, then their relative changes $\left(\frac{∆℘}{℘}and\frac{∆g}{g}\right)$ are proportional by an empirical factor, z: (3)$$\frac{∆℘}{℘}=z\frac{∆g}{g}$$

With the integration of (3), we obtain the following:(4)$$℘=qg^{z}$$

Importantly, (3) is a basic construction factor of all fractal structures when the template is repeated with different scales but the same form in a hierarchical order and changes the full structure with different magnifications. The structure of the biological structures follows the concept of the fractal geometry [230]. The metabolic power and its fluctuation have universal scaling, as well [231]. The scaling is shown for a large category of the living structures and processes [41]. This optimizing was rigorously formulated by the scaling idea and could be discussed in a universal frame, even on the subcellular energy consumption, as well as even the mitochondria and the respiratory complexes.

The cancer sequencing generations have a 1/f power-law distribution of mutant frequencies [232,233], showing a fingerprint of the self-organized behavior of the genetic organization. The possible way of the genetic self-organization may be basically connected to the organized bonds in DNA connected to the hydrogen bridges. These bridges form the nucleotide sequences in the DNA. In the connection of adenine (A) and thymine (T), the number of hydrogen bridges is $N_{H}=2$, while it is $N_{H}=3$ in the bonds of cytosine (C) and guanine (G). The combination of the genetic letters has a matrix presentation with Kronecker matrix multiplication [234]. The power of the Kronecker multiplication forms doublets, triplets, quadruplets, etc., of the combination of the A, T, C, and G genetic letters. (In RNA uracil (U) is mostly presented instead of T in DNA.) When the development of the life evolutionary varies all triplets randomly, it is enormously large: $64!\approx 10^{89}$. However, the character of the 20 amino acids that form the triplets of four nucleotides shows a symmetrical character, as the Kronecker matrix represents it [235], so the coding had definite rules, showing order in the large number of variants, dividing into two main, four sub, and eight sub-sub groups of these [236]. Due to the Markovian transitions, the introduced matrices are stochastic [237,238]. The higher power of the ${\left[\begin{array}{cc} C & T\\ A & G \end{array}\right]}^{n}$ matrix (could be named “n-plets” by generalization of triplets in ${\left[\begin{array}{cc} C & T\\ A & G \end{array}\right]}^{3}$), allows us to introduce the two-dimensional distances between them by the formal Euclidean distance [239].

The self-organizing and the consequent symmetry, which may deviate the evolutional processes from the complete randomness, may be connected in the energetical optimization of life and emphasize the main principles of the game above applied game considerations where the energy payoff had a central role in the changes. The directed energy-guided sequences of the cancer development solve the apparent contradiction in the expected linearity of the entirely random cancer prevalence. The limited random mutations of DNA replication are responsible for cancer. These mutations could be occasional (bad luck). They may be induced by various environmental changes, mutagen exposures, and inherited traits, which develop cancer during a long process of many repeated DNA replications, but some of the mutations are restricted by the presented genes in the connected gene family [223]. This limited randomness develops the cancer in sequences and nonlinearly drives its prevalence. The simple statistical variance concept does not reasonably describe the process [200]. The prevention of the development of mutated cells is more complex than averting exposure to mutagens from the environment throughout the organism’s lifetime. The already present mutations in the cellular population of the healthy organism can be long-term favorable, serving the evolutionary adaptation of the species by the selective pressure of the circumstances. The mutated cells in shorter timescales than the life span could serve without any clinical manifestation or may have apoptotic clearance from the system. A small number of the mutated cells survive multiple precancerous replications, collecting more mutations and adapting to the enormous stress, forcing the stroma for support, and resetting the complete homeostatic regulation. The development stages can be followed in the subclinical process until the disease has clinical symptoms. The subclinical stages accumulate the deemed harmless mutations, which induce accelerated DNA repair errors. The energy-optimizing evolutional game theory may describe all development stages. The complex Nash equilibrium with different optimums characterizes all the stages. The equilibrium maximizes the energetic payoff of individual cells, and, later, the micro-cluster of the tumor strives to enforce its interest; the micro-cluster is multilayered and combines the self-interest and local possibilities with the group interests [240], harmonizing the short-range and long-range interactions in the cellular structure.

The interplay between interspecies allometry, minimal entropy production, and homeostatic control highlights living systems’ intricate organization and adaptive strategies, including the deviations leading to cancer. Understanding these connections offers valuable insights into physiological optimization, how to achieve optimal function, and resource utilization. We may understand the mechanisms of cancer development, how the disruption of the optimizing principles (including the deviations in the homeostatic regulation) contribute to malignant conditions, and build a therapeutic strategy to restore homeostasis and minimize the entropy production in dysregulated conditions. The predictability of gene interactions could be a great addition to personalized medicine, with a better estimation of the risk of the malignancy, and increase the treatment efficacy.

## 5. Conclusions

The cancer development process may be divided into six stages. The first stage accumulates harmless, individually marginal mutations, modifying low-impact genes. The newly coming mutation “selfishly” tries to keep the cellular integrity, but many become apoptotic and do not survive the genetic modifications. The survival game is guided by Darwinian evolution, which selects some cellular replications that are strong enough to remain alive. In this way, adding many marginal mutations could be severe, and the process turns to the second stage, where one caretaker gene is mutated and causes genetic instability. Here, a new competitive game starts between the stroma and tumor cells. The winning cells turn to the third stage, where the gatekeeper gene is mutated and leaves the cellular proliferation uncontrolled, and it finally leaves the mutated cell in a unicellular setup with binary cooperation with the nearby stromal cell. In the fourth stage, the cancer cell adapts to challenging conditions and self-organized changes when it may enjoy the support of the healthy host. The new game is not competitive. It is cooperative, where the cancer cells commonly represent their demands. Cancerous clusters of cells have a survival advantage in the actual environmental circumstances. In the fifth stage, systemic physiological changes appear, developing the sixth stage, the clinical symptoms. All stages can be extended in time, depending on how long the Nash equilibrium exists in that stage.

The stability of the stages is sensitive; it is subject to permanent, almost random mutations, and the stages become unstable when they reach a harmful level of collected mutations. The apoptotic clearance has a high probability in every mutation step. The process may break in every phase of the development. On the other hand, the partly deterministically guided random mutations could develop “jumps”, and the clinical symptoms appear earlier than after six stages. This is observed in the broader range of the power function of the epidemiologic observations [12,13,14,16].

Furthermore, the mutations are not only deleterious. Random mutations are the starting points of the Darwinian selection of species, making the vast variation in the living species available. The derailing of cancer from normal evolution is when the genetic mutations are accumulated to such an extent that the cells start their particular adaptation mechanism in the organism and develop unicellular survival.

In summary, the extensive mutagen facilities in the standard circumstances (Figure 1) and the unprovoked random, but structurally limited mutations in the DNA replication may be arranged in six stages, appearing in the epidemiologic data. However, cancer development is a complex multistep process, and the statistics do not cover only six mutations but only six more or less differing stages, which may include an accumulation of many mutagen effects.

## Figures and Tables

**Figure 1 cells-13-00197-f001:**
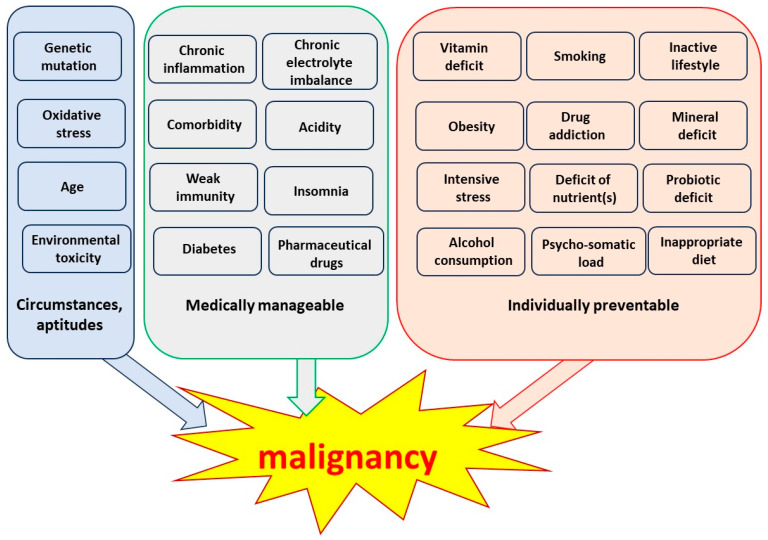
Some cancer-causing occurrences for humans. These are possibilities. All have a nonzero probability of cancer but are not sure to develop a malignancy. Having multiple factors rapidly increases the likelihood of cancerous diseases.

**Figure 2 cells-13-00197-f002:**
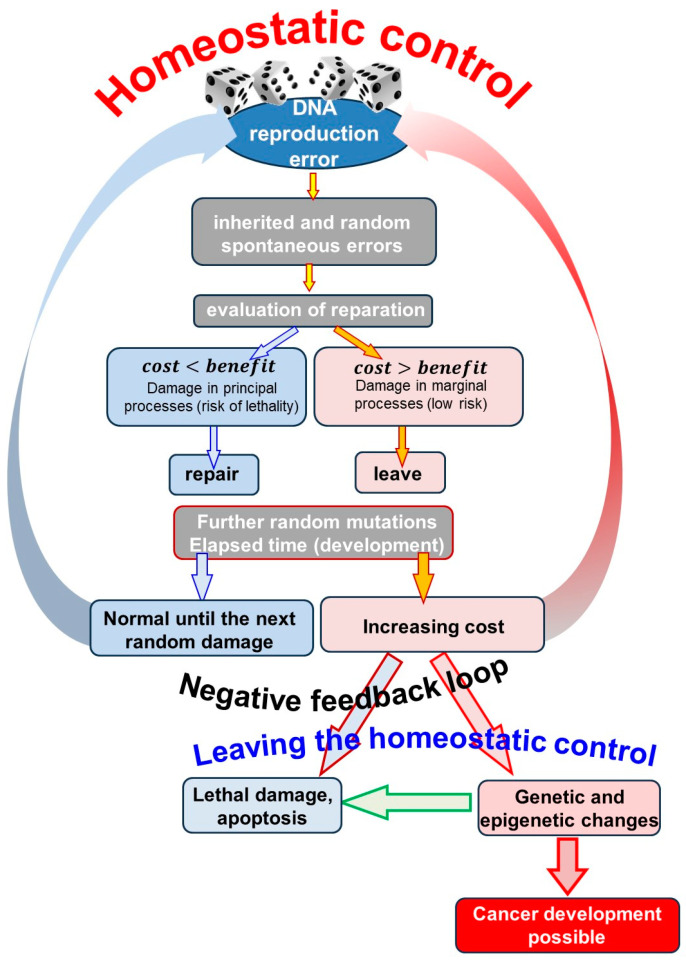
The homeostatic control loop under the permanent random mutations by the nonperfect DNA replication and the changing internal and external conditions. The development of the mutated genetic network needs an energetic decision to repair and spend energy, or it does not care about it because it is decided to be nonharmful. When the mutation could cause lethal damage, repair is necessary, and control is established. When judged as marginal, it was exposed to new mutations accumulated by cell cycles. It could cause apoptosis, which solves the problem, or it could go directly to the cancerous line.

**Figure 3 cells-13-00197-f003:**
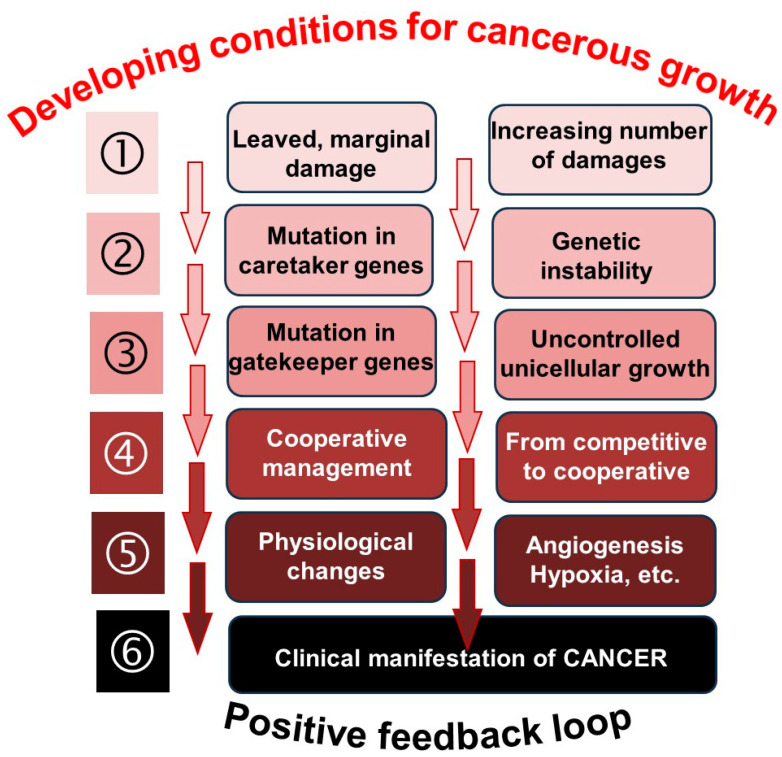
The six stages of cancer growth. The first stage produces precancerous cells by marginalizing some mutations during the repeated cell cycles. In the second stage, the caretaker genes are damaged. Their mutation causes genetic instability. The gatekeeper genes are mutated in the third stage, leaving the growth uncontrolled. The next stage changes the game. The cancer cells cooperatively adapt to the challenges, and the healthy cells support it. The central physiological networks are remodeled in the fifth stage to support malignant growth and invasion. In the last stage, the tumor is clinically manifested. The case could be epidemiologically registered when the cancer leaves the subclinical stages and appears in the clinical statistics.

**Figure 4 cells-13-00197-f004:**
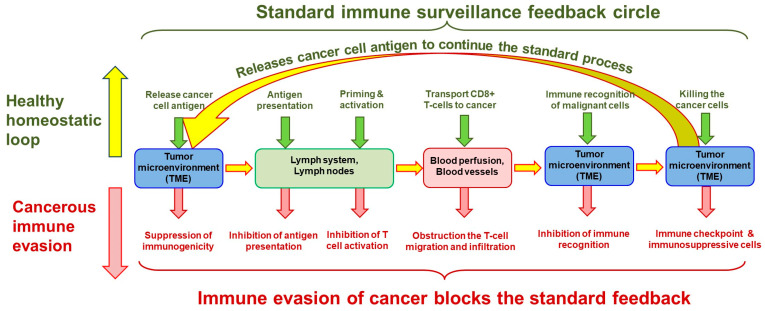
The malignancy evades immune surveillance. The standard immune control is diverted in all of its steps, causing tumor evasion.

**Figure 5 cells-13-00197-f005:**
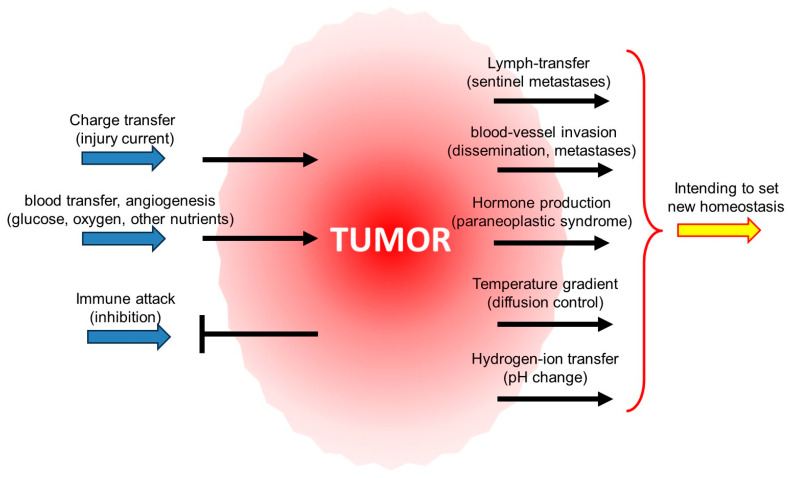
The physiological changes reshape homeostatic control. The charge (injury current), liquid (blood and lymph) transport, and immune inhibition set a condition to reset homeostasis. The consequences are the dissemination of the malignant cells, a particular hormone production, increased diffusional possibilities, and lowered pH. Note that the retuning of the body’s equilibrium supports the tumor-organs hypothesis and does not fit the atavistic explanations.

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
