# Peer review of "Peto’s “Paradox” and Six Degrees of Cancer Prevalence"

_cells, 2024, doi:10.3390/cells13020197_

Round 1

Reviewer 1 Report

Comments and Suggestions for Authors

In this Review, the Author provides analysis of the Peto's paradox and the six degrees of tumor prevalence through a theoretical lens. The Review integrates concepts of evolutionary biology and genetic mutations to propose a model explaining the non-linear progression of cancer development. The application of Darwinian natural selection to elucidate the role of mutations in the cell population of a healthy organism is a commendable approach. This furthers understanding of cancer development and progression beyond traditional statistical concepts. Recognizing that preventing the development of mutated cells is more difficult than avoiding mutagenic exposure demonstrates an understanding of the multifaceted nature of cancer etiology.  The Review presents an innovative perspective on cancer development that integrates evolutionary principles and proposes a new model.

This Review can be published after major revision.

The major concern is with the understanding of the Peto's paradox, which is the central point of this paper (that is reflected both in its title and as the aim of the study declared in the first sentence of Abstract). The Peto's paradox states that in the larger mammal species, the incidence of cancers is not higher than in the small mammals, which is surprising because larger species have more cell divisions,. The key word here is 'species'. Species, not specimens of one and the same species. The small species are selected for rapid reproduction and have the relaxed cell cycle control (10.1007/s00335-015-9605-8). The larger species are selected for longer life and reliability of organization; therefore they have the more stringent cell cycle control. In the larger species, the purifying selection is concentrated on the "information technology" of life, i.e., regulation of gene expression and development (10.1016/j.ygeno.2018.02.015). Increased accuracy of information processes allows them to escape "error catastrophes" in spite of the greater number of cell divisions and higher longevity (10.1016/j.ygeno.2018.02.015). This is the solution of the Peto's paradox.

The specimens of the same species, which greatly vary in body size (e.g., dogs), do not show Peto's paradox. In particular, osteosarcomas occur in large dogs 200 times more frequently than in small- and medium-sized breeds 10.1007/s00335-015-9605-8. This effect was also observed in humans, but to a lesser extent (10.1016/j.tree.2011.01.002 ), which is quite understandable, taking into account a lesser variation in body size. Moreover, across-tissues comparison within the human organism showed that the lifetime risk of cancer is strongly correlated with the total number of cell divisions in a given tissue (10.1126/science.1260825). Thus, there is no Peto's paradox in the humans. These facts confirm the explanation of the across-species Peto's paradox provided above.

In my view, the author should acknowledge this conceptual framework in the introduction to his model.

The small points:

1) In the Abstract, the fifth stage is repeated twice and there is no sixth stage (probably, a typo).

2) The Abstract would benefit from the conclusion explaining the importance and novelty of the presented concept for fundamental biology and clinical science.

Comments on the Quality of English Language

English can be improved in some places.  The Author can easily address this small flaw.

Reviewer 2 Report

Comments and Suggestions for Authors

Thank you for submitting this interesting review.  However it did not add significantly to the literature.  I would suggest to add the following sections: The therapeutic implications and novel cancer prevention's tools as well as the challenges and limitations of the study. 

Comments on the Quality of English Language

English language was fine however the review needs extensive editing and typo corrections

Round 2

Reviewer 1 Report

Comments and Suggestions for Authors The author has done a great job and
significantly improved the article. It can be published.

Reviewer 2 Report

Comments and Suggestions for Authors

Thank you for submitting this revised manuscript which has been significantly improved.